# OpenReview forum: "TIS-DPO: Token-level Importance Sampling for Direct Preference Optimization With Estimated Weights"
_ICLR.cc/2025/Conference — ICLR 2025 Poster_

### Official Review · Reviewer_pMc1 · 2024-10-30

**Soundness:** 2
**Presentation:** 2
**Contribution:** 2
**Rating:** 5
**Confidence:** 4

**Summary:**

The paper proposes TIS-DPO (Token-level Importance Sampling for Direct Preference Optimization), which is built upon standard DPO by assuming that different tokens within a response have varying levels of importance. Specifically, the importance of tokens in the completion is estimated based on 2 separate reward models. The authors demonstrate that on two datasets the proposed method  outperforms baseline methods.

**Strengths:**

- The paper proposes a new perspective of preference learning based on DPO by considering token-level importance.
- Nice appendix.
- The experimental evaluation is comprehensive, spanning 2 datasets, different metrics including MT-bench and win rates, and multiple baselines (DPO, PPO, IPO, TDPO, KTO) .

**Weaknesses:**

- Questionable Motivation. Even though we assume you can accurately estimate the token importance, the token-level DPO approach seems misaligned with how preference data is collected, where annotators make holistic judgments about complete responses, not token-by-token evaluations.
- The accuracy and reliability of token-level importance estimation are not thoroughly validated
- The approach assumes tokens can be evaluated independently, which ignores crucial context dependencies in language.
- Introducing reward models for token importance estimation seems to contradict DPO's core advantage of avoiding reward modeling. Should we compare against PPO, where the rewards are estimated?
 - The additional cost of training and maintaining contrastive LLMs for weight estimation is not analyzed. No discussion of the computational trade-offs compared to standard DPO. The scalability implications for larger models are not addressed
-  Introduction of new hyperparameter μ adds complexity. Limited analysis of hyperparameter sensitivity. No clear guidance for hyperparameter selection in different scenarios
- Missing analysis of how sequence length affects performance. No investigation of the relationship between token position and estimated importance.
- Table 2 presentation should have been corrected.

**Questions:**

- Why is the DPO result missing in Table 1?
- How many LLMs are needed in your training process? Is it 3?
- Will your proposed method be sensitive to sequence length in better or worse completions, where longer sequences may have higher rewards?
- Regarding the win rate, I understand that GPT-4 is the evaluator model - which model is being used as the competitor?
- In Table 1, is there any reason why the win rate increases significantly while the MT-bench score is not improved comparably?

---

> ### Author Response · Authors · 2024-11-20
> **Thanks for your careful review (Part 1/3)**
>
> ## **1. About Motivation**
>
> > Questionable Motivation. Even though we assume you can accurately estimate the token importance, the token-level DPO approach seems misaligned with how preference data is collected, where annotators make holistic judgments about complete responses, not token-by-token evaluations.
>
> Thank you for raising this important point. While annotators indeed make holistic judgments rather than token-by-token evaluations, there are two key factors that support our motivation:
>
> 1. In any given response, different tokens **naturally have varying levels of importance**, regardless of whether annotators evaluate token-by-token or holistically. This can be clearly seen in Figures 4-7 of our paper, which demonstrate how different tokens contribute differently to the overall preference.
>
> 2. DPO optimization fundamentally works by **increasing the probability of positive responses** while **decreasing the probability of negative ones**. Since a response's probability is the product of its **token probabilities**, DPO essentially **increases the probability of tokens** in winning responses while decreasing those in losing responses. However, even when a winning response is holistically better than a losing one (based on annotator judgments), **some tokens in the winning response may not deserve probability increases** (for example, in safety alignment, tokens that are irrelevant to safety or don't actually improve safety). Understanding this makes our motivation clear - we aim to perform **more fine-grained DPO optimization** at the token level through importance estimation.
>
> Although annotators do not explicitly label token-level importance, this information can be learned implicitly from large amounts of response-level annotations. This is precisely what Rafailov et al. (2024a)[1] discovered - models trained with DPO exhibit a degree of token-level interpretability. We leverage this property in TIS-DPO(D) to uncover the token-level importance embedded within annotator judgments, allowing us to more precisely determine which token probabilities should be increased or decreased during optimization.
>
> [1] From r to Q∗: Your Language Model is Secretly a Q-Function
>
> ## **2. About evaluation of token-level importance**
>
> > The accuracy and reliability of token-level importance estimation are not thoroughly validated
>
> Please refer to the **fourth** response in **response to all reviewers**.
>
> ## **3. About independent token evaluation**
>
> > The approach assumes tokens can be evaluated independently, which ignores crucial context dependencies in language.
>
> We believe there may be some misunderstanding regarding this concern. We do not assume **tokens can be evaluated independently**. In fact, when estimating token importance using the formula $\log\frac{\pi^{+}(y_t \mid x, y^{<t})}{\pi^{-}(y_t \mid x, y^{<t})}$, we explicitly consider the context $x,y^{<t}$, which means we **take into account the context dependencies** in language.
>
> Furthermore, we would like to clarify that having different token importance weights is **distinct** from evaluating tokens independently.
>
> We simply believe that in the context of each token, different importance weights should be assigned during the DPO optimization process.
>
> ## **4. About computational cost**
>
> > The additional cost of training and maintaining contrastive LLMs for weight estimation is not analyzed. No discussion of the computational trade-offs compared to standard DPO. The scalability implications for larger models are not addressed
>
> Please refer to the **first** point in the **response to all reviewers**.
>
> ## **5. Hyperparameter Analysis**
>
> > Introduction of new hyperparameter μ adds complexity. Limited analysis of hyperparameter sensitivity. No clear guidance for hyperparameter selection in different scenarios
>
> Please refer to the **second** point in the **response to all reviewers**.

---

> ### Author Response · Authors · 2024-11-20
> **Thanks for your careful review (Part 2/3)**
>
> ## **6. About Comparing to PPO**
>
> > Introducing reward models for token importance estimation seems to contradict DPO's core advantage of avoiding reward modeling. Should we compare against PPO, where the rewards are estimated?
>
> First, we need to clarify that we did not fully introduce a reward model - we only used contrastive LLMs to estimate token weights, which cannot be considered explicit reward modeling.
>
> Furthermore, we believe DPO's advantages are not just about avoiding reward modeling. More importantly, it uses off-policy data to avoid reinforcement learning during training to improve efficiency. Our method maintains this advantage, only adding pre-computed token weights to the training objective without any additional computational overhead during DPO optimization.
>
> Most importantly, even PPO's optimization approach cannot flexibly determine weights for each token during training. Current PPO methods determine rewards for the entire sequence and cannot precisely assign rewards to individual tokens. For example, in PPO, the token-level rewards are typically generated only after the final token is complete, updating based on the entire sequence's reward, as shown in the following formula [1]:
>
>
> $r(\mathbf{s}_t, \mathbf{a}_t)$ is defined as:
>
> When $\mathbf{s}_{t+1}$ is not terminal:
> $r(\mathbf{s}_t, \mathbf{a}_t) = \beta \log \pi(\mathbf{a}_t|\mathbf{s}_t)$
>
> When $\mathbf{s}_{t+1} = \mathbf{y}$ is terminal:
> $r(\mathbf{s}_t, \mathbf{a}_t) = r(\mathbf{x}, \mathbf{y}) + \beta \log \pi(\mathbf{a}_t|\mathbf{s}_t)$
>
>
> Our approach, by pre-estimating weights for each token and optimizing more accurately during training based on these token-specific weights, offers an advantage that neither previous PPO nor DPO methods possess.
>
> Additionally, as shown in Table 1, our method has been compared with PPO and demonstrates superior performance.
>
> [1] From r to Q∗: Your Language Model is Secretly a Q-Function
>
>
>
>
> ## **7. About length impact**
>
> > Missing analysis of how sequence length affects performance. No investigation of the relationship between token position and estimated importance.
>
> Thank you for your suggestion. After analyzing the average weights at each position, we indeed found that in both chosen and rejected responses, there is a trend where weight values increase with position when the sequence length is long enough. The most likely reason for this is that we used contrastive DPO-trained LLMs to help estimate weights, and DPO tends to be influenced by sequence length. We have updated the figure in the appendix K.
>
> Based on this observation, we conducted a simple experiment by adding a weight decay factor $\lambda$, where the weight decay for position N is $\lambda^{N-1}$. We found that this weight decay mechanism led to slight improvements in performance.
>
> | Method | Position Weight | Llama-Guard ↑ | Harm. ↓ | Help. ↑ | MT ↑ | Win ↑ |
> |---------|-----------------|---------------|----------|----------|-------|--------|
> | TIS-DPO(D) | Original | 96.7% | 0.1 | 8.0 | 4.3 | 79.3% |
> | TIS-DPO(D) | With Decay (0.995) | 97.9% | -0.3 | 8.1 | 4.4 | 79.9% |
>
> We have included a more detailed discussion of this issue in the appendix of our updated paper. We greatly appreciate your insightful suggestion, which has significantly contributed to improving the quality of our paper.
>
> **We have added the related discussion to the appendix K of our updated paper (highlighted in red).**
>
>
> ## **8. Table 2 presentation**
>
> Thank you for pointing this out. We have revised the layout of Table 2 accordingly.
>
> ## **9. About the GPT-4 evaluation**
>
> > Why is the DPO result missing in Table 1?
>
> Thank you for your careful observation. This is because all win rates in Table 1 use DPO output results as the baseline, as we mentioned in line 323 of the original paper.
>
> > Regarding the win rate, I understand that GPT-4 is the evaluator model - which model is being used as the competitor?
>
> This question is related to the one above - the competitor is the DPO model.  Therefore, we did not list the win rate for the DPO model.
>
> ## **10. How many LLMs are needed in your training process? Is it 3?**
>
> For TIS-DPO(P), only one LLM needs to be trained throughout the entire process. (the number is 1)
>
> For TIS-DPO(S/D), we first need to train two contrastive LLMs, then use these two LLMs to annotate data, and finally train TIS-DPO. (the number is 3)
>
> However, in the final TIS-DPO training process, once we have the token-level importance, only one LLM is needed for training.

---

> ### Author Response · Authors · 2024-11-20
> **Thanks for your careful review (Part 3/3)**
>
> ## **11. Will your proposed method be sensitive to sequence length in better or worse completions, where longer sequences may have higher rewards?**
>
> To be honest, we are not entirely clear about the meaning of this question. What we can easily verify is whether outputs of different lengths from our model show different reward performance. Specifically, we used the PKU-Safety dataset and the beaver-cost-3.0 reward model for verification. Here are our comparison results:
>
> | Method | Length Range | Average Cost Value (↓) | Sample Count |
> |---------|--------------|------------------|--------------|
> | Llama2-7b-SFT | 0-49 | 7.764 | 123 |
> | Llama2-7b-SFT | 50-99 | 7.994 | 486 |
> | Llama2-7b-SFT | 100-149 | 8.286 | 299 |
> | Llama2-7b-SFT | 150-199 | 8.296 | 91 |
> | Llama2-7b-DPO | 0-49 | 3.930 | 85 |
> | Llama2-7b-DPO | 50-99 | 3.764 | 345 |
> | Llama2-7b-DPO | 100-149 | 4.352 | 416 |
> | Llama2-7b-DPO | 150-199 | 4.141 | 154 |
> | Llama2-7b-TIS-DPO (Ours) | 0-49 | 0.631 | 67 |
> | Llama2-7b-TIS-DPO (Ours) | 50-99 | 0.165 | 196 |
> | Llama2-7b-TIS-DPO (Ours) | 100-149 | -0.030 | 338 |
> | Llama2-7b-TIS-DPO (Ours) | 150-199 | -0.656 | 398 |
>
> From the results above, we observe an interesting phenomenon: With Llama2-7b as the base model, the output quality decreases with increasing length for models with only SFT training. DPO also shows a declining trend with increasing length. However, our TIS-DPO actually shows improving output quality (safety) as length increases, indicating that our model is not sensitive to length increases. We also found that models trained with TIS-DPO tend to generate slightly longer texts.
>
> Thank you for bringing this up. We have added these experiments to the appendix of our revised version.
>
> Additionally, your question might also be referring to sensitivity to length distribution in the dataset. Since we train and compare with DPO on the same dataset, and our method outperforms DPO across different output lengths, this potentially indicates that our method is not particularly sensitive to training data length, at least not more so than DPO.
>
> ## **12. In Table 1, is there any reason why the win rate increases significantly while the MT-bench score is not improved comparably?**
>
> We can explain this from two perspectives - one is the numerical reason, and the other is the underlying mechanism:
>
> **numerical reason**
>
> This is because our win rate is evaluated on generated data from the test sets of the corresponding datasets (PKU-Safety and Anthropic-HH). Both improvements in harmlessness and helpfulness can increase the win rate in these datasets. In our training, the improvement in safety is more significant, which leads to a larger increase in win rate.  And this does not mean overfitting, because we also tested safety on external datasets like Advbench.
>
> Meanwhile, MT-bench is a third-party dataset with a different data source from PKU-Safety and Anthropic-HH. Since it focuses primarily on helpfulness, reasoning, and coding abilities, the final improvement on MT-bench is not as significant.
>
> **underlying mechanism**
>
> We found that if the performance gap between the contrastive LLMs (positive and negative LLMs) is large enough, the accuracy of token importance estimation will be higher, and the performance gain will be larger. In Table 1, we observe that the performance gap between the trained contrastive LLMs in safety is larger, but the performance gap in helpfulness and MT-bench is not as significant. This may be the underlying reason.

---

> ### Author Response · Authors · 2024-11-27
>
> Dear reviewer pMc1,
>
> We appreciate your careful review of our paper, and we believe we have provided thorough responses. Would you be able to give us further feedback?
>
> Best regards

---

### Official Review · Reviewer_UoKq · 2024-11-03

**Soundness:** 2
**Presentation:** 3
**Contribution:** 2
**Rating:** 5
**Confidence:** 4

**Summary:**

This paper introduces a new approach, Token-level Importance Sampling DPO (TIS-DPO), to improve the efficiency and effectiveness of Direct Preference Optimization (DPO) for Large Language Models (LLMs). The authors propose that optimal data for DPO should have equal expected rewards for each token in winning and losing responses. They use importance sampling with the original dataset to achieve unbiased optimization and estimate token importance weights using the difference in prediction probabilities from contrastive LLMs. Experiments demonstrate that TIS-DPO outperforms baseline methods in harmlessness, helpfulness alignment, and summarization tasks.

**Strengths:**

1. The paper presents a clear and well-structured method, TIS-DPO, which addresses a limitation of traditional DPO by considering the importance of individual tokens.

2. The experimental results show improvements over various baselines, indicating the practical value of the proposed method.

**Weaknesses:**

1. Lack of In-depth Analysis: The paper lacks a detailed analysis of why the three proposed methods for constructing contrastive LLMs can effectively measure token importance during training. Additionally, there is no detailed explanation for the significant differences in performance among these methods.

2. Fixed Token Importance: The token importance weights are generated before training and remain fixed throughout the process. However, as the model evolves during training, the importance of tokens may also change, which could affect the effectiveness of the method.

3. Scalability and Generalizability: The paper does not thoroughly address the scalability and generalizability of the proposed method. It is unclear whether every model would require its own set of token importance weights or if the one set of weights can be generalized to different models and tasks.

4. Limited Experimental Scope: While the method shows effectiveness in improving harmlessness and helpfulness, the experiments do not cover other critical areas such as code and mathematics, where token-level importance might be even more crucial. Further experiments in these domains would strengthen the validation of the method.

**Questions:**

Please refer to the Weaknesses.

---

> ### Author Response · Authors · 2024-11-20
> **Thanks for your careful review**
>
> ## **1. About In-depth Analysis**
>
> > Lack of In-depth Analysis: The paper lacks a detailed analysis of why the three proposed methods for constructing contrastive LLMs can effectively measure token importance during training. Additionally, there is no detailed explanation for the significant differences in performance among these methods.
>
> Thank you for your suggestion. We need to clarify that we do not measure token importance during training. Instead, after obtaining the contrastive LLMs, we estimate the reward $\log\frac{\pi^{+}(y_t \mid x, y^{<t})}{\pi^{-}(y_t \mid x, y^{<t})}$ by comparing the actual probability differences between these contrastive LLMs on each token, and then derive our importance weights.
>
> Theoretically, early research has found that the difference between DPO-trained models and the original LLM can be used to estimate rewards[1]. Additionally, in Theorem 2, we proved that token importance correlates with reward values. Therefore, we can use the differences between contrastive LLMs to estimate importance weights.
>
> Furthermore, we acknowledge that we should conduct more detailed analysis of the token importance estimation methods.
>
> Since other reviewers raised similar concerns, we addressed this issue in **fourth point of our response to all reviewers** (also in Appendix I). The analysis concluded that TIS-DPO(D) provides the best importance weight estimation while TIS-DPO(P) performs the worst. This effectively explains the performance differences between these three methods - better importance weight estimation leads to better training results compared to standard DPO.
>
> [1] From r to Q∗: Your Language Model is Secretly a Q-Function
>
> ## **2. About fixed weights**
>
> > Fixed Token Importance: The token importance weights are generated before training and remain fixed throughout the process. However, as the model evolves during training, the importance of tokens may also change, which could affect the effectiveness of the method.
>
>
> We believe that the importance weights should only be related to the target (harmlessness, helpfulness, etc.) and not the specific LLM itself. Therefore, from this perspective, if we can predict a perfect importance weights, it is unnecessary to adjust them during training.
>
> We admit that different LLMs may have varying accuracy in estimating importance weights.
>
> However, it's worth noting that the goal of this paper is not to propose the most accurate token importance weights, but rather to demonstrate through theory and experiments that introducing importance weights can improve DPO training effectiveness. Future work can focus on predicting better importance weights.
>
>
> ## **3. About Generalizability**
>
> > Scalability and Generalizability: The paper does not thoroughly address the scalability and generalizability of the proposed method. It is unclear whether every model would require its own set of token importance weights or if the one set of weights can be generalized to different models and tasks.
>
> Thank you for your suggestion. Theoretically, our importance weights are LLM-agnostic and should generalize across all LLMs. However, in practice, one challenge is that different LLMs use different tokenizers, making it difficult to perfectly align weights. To investigate this, we conducted experiments using two LLMs with identical tokenizers - Llama2-7B and Llama2-13B. We found that using importance weights generated by Llama2-13B to train Llama2-7B achieved excellent results, even slightly outperforming the weights estimated by Llama2-7B itself.
>
> | Method | Llama-Guard ↑ | Harm. ↓ | Help. ↑ | MT ↑ | Win ↑ |
> |---------|---------------|----------|----------|-------|--------|
> | TIS-DPO(D) 7B→7B | 96.7% | 0.1 | 8.0 | 4.3 | 79.3% |
> | TIS-DPO(D) 13B→7B | 97.1% | 0.1 | 8.1 | 4.4 | 80.5% |
>
> These experimental results demonstrate that our token importance weights have good generalizability across different model sizes.
>
> ## **4. About Experimental Scope**
>
> > Limited Experimental Scope: While the method shows effectiveness in improving harmlessness and helpfulness, the experiments do not cover other critical areas such as code and mathematics, where token-level importance might be even more crucial. Further experiments in these domains would strengthen the validation of the method.
>
> Please refer to the **third** point of our **response to all reviewers**.
>
> In summary, our experiments on the Ultrafeedback dataset (contains code and math) demonstrated significant performance improvements in MT-bench (contains code and math testing), which shows the general effectiveness of our method across different domains.

---

> ### Author Response · Authors · 2024-11-27
>
> Dear reviewer UoKq,
>
> We appreciate your careful review of our paper, and we believe we have provided thorough responses. Would you be able to give us further feedback?
>
> Best regards

---

### Official Review · Reviewer_D9Hy · 2024-11-03

**Soundness:** 3
**Presentation:** 3
**Contribution:** 4
**Rating:** 10
**Confidence:** 5

**Summary:**

The paper makes a key claim that "the most stable form of DPO loss occurs when tokens in winning and losing responses have identical expected rewards." And the best way to estimate the unavailable ideal DPO dataset is to use importance sampling. The paper did a thorough proof combining prior works of Zeng et al. to show a token-level DPO objective, and then enhance it with importance sampling to assign importance weight on each token of actual data distribution to approximate the data distribution of ideal equal importance DPO dataset.

It then proposes 3 ways to estimate the token-wise weighting by contrasting LLM likelihood of 1. a prompt-based method. 2 SFT two models to desirable and undesirable. 3. DPO two models to desirable and undesirable. Experiment results show 1 doesn't work, and 3 works surprisingly well, significantly exceeding sequence level DPO baseline on HH, PKUSafe, and Summarization dataset.

**Strengths:**

* Very novel idea of using contrastive LLM likelihood as token-level weights for importance sampling.

* Very important area of research to improve the popular off-line alignment DPO algorithm.

* Great insight about limitation of sequence level DPO, thus introducing the need for nuanced token-level optimization

* Surprisingly good improvements against DPO baseline using token-level importance sampling based TIS-DPO.

* Very thorough proof and derivation process for the TIS-DPO objective

**Weaknesses:**

* “Although Rafailov et al. (2024a) demonstrate that DPO possesses a certain degree of token-level interpretability,” It would be helpful to point to the exact section or figure of this analysis.

* The strength of the method could benefit from more thorough experiment on stronger alignment datasets. HH and Summarization are commonly used but relatively poor quality data, compared with some newer alignment datasets like Ultrafeedback and HelpSteer2.

* It is clear that TIS-DPO has significant strength in aligning models to safety and slight improvement to helpfulness. I am curious to see the dynamic of using TIS-DPO to align stronger models like Gemma2-9B or Llama-3-8B.

Overall, great paper, excellent intuition, and solid experiment results.

**Questions:**

Could you explain why or provide some intuition as to why the the improvements is much better for safety alignment than helpful alignment?

Also, it would be great to have some empirical experiments to see whether TIS-DPO is leading to faster convergence (which may be  case based on the limited experiment setup shown in paper) or in fact better convergence, leading to meaningfully better aligned models. Could you expand on this, and how you may design experiments to illustrate it?

---

> ### Author Response · Authors · 2024-11-20
> **Thanks for your careful review**
>
> First, we sincerely thank you for your high recognition of our paper. We are truly honored, and we will now address your questions below.
>
>
> ## **1. About token-level interpretability in DPO**
>
> > “Although Rafailov et al. (2024a) demonstrate that DPO possesses a certain degree of token-level interpretability,” It would be helpful to point to the exact section or figure of this analysis.
>
> This topic is discussed in detail in Section 4.2 of Rafailov et al. (2024a) [1]. Thank you for bringing this to our attention - we have included the relevant description in the updated version of our paper.
>
> [1] From r to Q∗: Your Language Model is Secretly a Q-Function
>
> ## **2. Use more advanced model**
>
> > It is clear that TIS-DPO has significant strength in aligning models to safety and slight improvement to helpfulness. I am curious to see the dynamic of using TIS-DPO to align stronger models like Gemma2-9B or Llama-3-8B.
>
> Thank you for your suggestion. We agree that experimenting with more powerful LLMs is essential, so we conducted experiments with Llama-3-8B. Due to time constraints, we haven't completed full experiments with Gemma2-9B yet, but we believe our current results already demonstrate our point (our method could work on stronger LLMs).
>
> | Method | Llama-Guard ↑ | Harm. ↓ | Help. ↑ | MT ↑ | Win ↑ |
> |---------|---------------|----------|----------|-------|--------|
> | DPO | 88.3% | 3.9 | 8.4 | 5.5 | - |
> | TDPO | 91.1% | 3.6 | 8.5 | 5.6 | 53.2% |
> | TIS-DPO(S) | 94.8% | 2.1 | 8.6 | 5.7 | 68.2% |
> | TIS-DPO(D) | 98.9% | 0.0 | 8.9 | 6.2 | 71.5% |
>
> In the Llama3-8B experiments, we observe that since Llama3 has better inherent safety than Llama2, the relative improvement in safety metrics is smaller. However, we see slightly larger improvements in helpfulness and MT-bench scores.
>
> ## **3. About experiment on new dataset**
>
> > The strength of the method could benefit from more thorough experiment on stronger alignment datasets. HH and Summarization are commonly used but relatively poor quality data, compared with some newer alignment datasets like Ultrafeedback and HelpSteer2.
>
> Please refer to the **third** response in **response to all reviewers**.
>
> ## **4. About intuition**
>
> > Could you explain why or provide some intuition as to why the the improvements is much better for safety alignment than helpful alignment?
>
> We can explain this from two perspectives:
>
> First, for safety-related tests, token importance is easier to determine because safety-related words are more readily identifiable. For other scenarios, identifying the importance of relevant words may not be as straightforward.
>
> The second important perspective is that our estimation of importance correlates positively with the **performance gap** between positive and negative LLMs. For safety-related scenarios, there is inherently a large performance difference between positive and negative models, which leads to better importance estimation and consequently better TIS-DPO performance. However, we observe that in helpfulness scenarios, the performance gap between positive and negative DPO training is not as significant, possibly due to **greater annotation noise** in PKU-safeRLHF and HH datasets regarding helpfulness.
>
> As shown in our supplementary experiments using the Ultrafeedback dataset above (the **third** response in **response to all reviewers**), when there is a larger performance gap between positive and negative LLMs, our method can achieve corresponding improvements.
>
> ## **5. About the faster convergence**
>
> > Also, it would be great to have some empirical experiments to see whether TIS-DPO is leading to faster convergence (which may be case based on the limited experiment setup shown in paper) or in fact better convergence, leading to meaningfully better aligned models. Could you expand on this, and how you may design experiments to illustrate it?
>
> We believe this experiment is relatively straightforward to design. By comparing the model performance between DPO and TIS-DPO at the same number of steps, we can determine whether TIS-DPO achieves faster convergence.
>
> Specifically, here are our experimental results:
>
> | Settings        | Llama-Guard ↑ | Harm. ↓ | Help. ↑ | MT ↑ |
> |----------------|---------------|----------|----------|-------|
> | DPO (1 epoch) | 74.4%         | 5.6      | 7.9      | 4.1   |
> | DPO (2 epochs) | 77.4%         | 4.9      | 8.0      | 4.2   |
> | DPO (3 epochs) | 79.8%         | 4.6      | 8.0      | 4.2   |
> | TIS-DPO(D) (1 epoch)    | 96.7%  | 0.1      | 8.1      | 4.3   |
> | TIS-DPO(D) (2 epochs)    | 97.1%  | 0.0      | 8.1      | 4.3   |
> | TIS-DPO(D) (3 epochs)    | 97.4%  | -0.1      | 8.1      | 4.3   |
>
> Based on these evaluation results, our method indeed shows better convergence properties compared to DPO, with smaller performance gaps between different epochs. Moreover, TIS-DPO achieves better performance in just one epoch than DPO does in three epochs.

---

> > ### Comment · Reviewer_D9Hy · 2024-11-25
> >
> > Thank you for your detailed response. The new result on Llama-3 is consistent with previous result reported in paper.
> > I think the idea of paper of using contrastive token probs for importance sampling is very interesting, and worth publishing to invite more interest and discussion.

---

> > > ### Author Response · Authors · 2024-11-25
> > >
> > > Thanks for your positive feedback. We appreciate your time and consideration in reviewing the manuscript.

---

### Official Review · Reviewer_SJB6 · 2024-11-03

**Soundness:** 3
**Presentation:** 4
**Contribution:** 3
**Rating:** 8
**Confidence:** 4

**Summary:**

The paper presents TIS-DPO (Token-Level Importance Sampling for Direct Preference Optimization), a novel approach designed to enhance the optimization of Large Language Models by assignig different weights to tokens based on their predicted rewards. It aims to achieve unbiased optimization by approximating an ideal dataset where each token has equal expected rewards.

Rely on the previous work TDPO (Token-level Direct Preference Optimization), this method estimates token importance weights using differences in prediction probabilities between pairs of contrastive LLMs. Three approaches are explored for constructing these contrastive models:
- Guiding the original LLM with contrastive prompts.
- Training separate LLMs on winning and losing responses.
- Performing forward and reverse DPO training.

Extensive experiments are conducted to evaluate TIS-DPO on tasks related to harmlessness, helpfulness, and summarization. Results indicate that TIS-DPO significantly outperforms several baseline methods, demonstrating improvements in both safety and response quality.

**Strengths:**

- TIS-DPO introduces a token-level importance sampling method that addresses the limitations of treating all tokens equally in Direct Preference Optimization (DPO), enhancing optimization efficiency.
- The proposed method is evaluated across multiple datasets, showing its robustness and versatility in different contexts and tasks.
- By using actual dataset with importance sampling instead of taking that dataset as an ideal dataset, the method is more practical for real-world applications where optimal data distribution is typically unavailable.

**Weaknesses:**

- The proposed method involves steps for constructing contrastive LLMs and tokeniwse importance weights estimating, which increases computational costs and training time compared to simpler baseline methods.
- The Hoeffding inequality is applied in the proof of Theory 1, which requires that the summation variables (i.e. token rewards) be i.i.d., but tokens in a sentence dose not conform to the condition of being i.i.d. along with their observed rewards.
- TIS-DPO(S/D) distributes the sentence "value" to the token level and in addition estimates token importance via reward evaluator driven by the same sentence "value". The potential cost is that the method may be more sensitive to the noise in data annotation. For example, if there is a certain proportion of noise in the training set, let's assume that some of the triples $(x, y_w, y_l)$ are mislabeled in reverse order as $(x, y_l, y_w)$ which are common in the real world. Then TIS-DPO(S/D) may be more affected than DPO. Can you add 20% of noise (i.e. swapping the chosen and reject of a triple) into the training data and compare the performance degradation of TIS-DPO(S/D) versus DPO?

**Questions:**

1. TIS-DPO(D) in Table 1 shows a significant improvement in performance far exceeds expectations, which seems a probable risk of overfitting. Can you conduct cross-validation to prove that there is no overfitting?
2. The experimental results of other methods on the Anthropic-HH dataset in Table 1 were not found in the cited papers, such as TDPO. Were all the comparative experiments were reproduced?
3. As quite a few hyperparameters have been introduced on top of DPO and TDPO, are there any sensitivity test on the newly added hyperparameters $k$, $\mu$, $L$ and $U$ in Eq. 19?

P.S.
- Zeng2024a and Zeng2024b are different versions of the same paper.
- Conflict usage of $\mu$ in Theorem 2, eq. 14 and eq. 19.

---

> ### Author Response · Authors · 2024-11-20
> **Thanks for your careful review (Part 1/3)**
>
> ## **1. About the computational cost**
>
> > The proposed method involves steps for constructing contrastive LLMs and tokeniwse importance weights estimating, which increases computational costs and training time compared to simpler baseline methods.
>
> Please refer to the **first** response in **response to all reviewers**.
>
> ## **2. About the Hoeffding Inequality**
>
> >The Hoeffding inequality is applied in the proof of Theory 1, which requires that the summation variables (i.e. token rewards) be i.i.d., but tokens in a sentence dose not conform to the condition of being i.i.d. along with their observed rewards.
>
> Thank you for your careful review. We agree that the Hoeffding inequality is not very suitable in our setting, and indeed the assumption that each token's reward is independent is inappropriate.
>
> Since we only aim to reach the conclusion that "the greater the difference in average rewards between the winning and losing responses, the higher the noise in the data and the less stable the optimization", which does not rely on Hoeffding inequality, we have re-derived our proof using **McDiarmid's inequality** (https://en.wikipedia.org/wiki/McDiarmid%27s_inequality) under more relaxed assumptions to reach the same conclusion.
>
> Under McDiarmid's inequality, we replace the formula in the Theorem 1 with the following, which still leads to the same conclusion.
>
> $$
> P(S_w \leq S_l) \leq \exp\left(-\frac{2(\mathbb{E}[S_w] - \mathbb{E}[S_l])^2}{\sum_{i=1}^{n_w} c_{w,i}^2/n_w^2 + \sum_{j=1}^{n_l} c_{l,j}^2/n_l^2}\right),
> $$
> where $c_{w,i} = b_w - a_w$ and $c_{l,j} = b_l - a_l$ are the maximum changes in the reward when modifying a single token, and $P(S_w \leq S_l)$ represents the probability of data noise.
>
> Below we provide the detailed derivation:
>
> First, we define our function of interest as the difference between the average rewards. This function captures the key quantity we want to analyze - the difference between winning and losing responses:
>
> $$f(r_{w,1}, \ldots, r_{w,n_w}, r_{l,1}, \ldots, r_{l,n_l}) = S_w - S_l = \frac{1}{n_w}\sum_{i=1}^{n_w} r_{w,i} - \frac{1}{n_l}\sum_{j=1}^{n_l} r_{l,j}$$
>
> To apply McDiarmid's inequality, we need to determine how much our function can change when modifying a single variable while keeping all others fixed. This helps us establish the bounded difference condition required by the inequality.
>
> For a winning token reward $r_{w,i}$ changed to $r_{w,i}'$, we can bound the change as follows:
>
> $$
> |f(\ldots, r_{w,i}, \ldots) - f(\ldots, r_{w,i}', \ldots)| = |\frac{r_{w,i}}{n_w} - \frac{r_{w,i}'}{n_w}| \\
> = |\frac{r_{w,i} - r_{w,i}'}{n_w}| \\
> \leq \frac{|b_w - a_w|}{n_w} = \frac{c_{w,i}}{n_w}
> $$
>
> This bound holds because both $r_{w,i}$ and $r_{w,i}'$ must lie in the interval $[a_w, b_w]$, so their maximum difference is $b_w - a_w$.
>
> Similarly, for a losing token reward $r_{l,j}$ changed to $r_{l,j}'$, we can establish:
>
> $$|f(\ldots, r_{l,j}, \ldots) - f(\ldots, r_{l,j}', \ldots)| \leq \frac{|b_l - a_l|}{n_l} = \frac{c_{l,j}}{n_l}$$
>
> Now we can apply McDiarmid's inequality. For a function where changing one input variable can change the output by at most $c_i$, McDiarmid's inequality states:
>
> $$\mathbb{P}(f - \mathbb{E}[f] \leq -t) \leq \exp\left(-\frac{2t^2}{\sum c_i^2}\right)$$
>
> Applying this to our case with our established bounds:
>
> $$\mathbb{P}(f - \mathbb{E}[f] \leq -t) \leq \exp\left(-\frac{2t^2}{\sum_{i=1}^{n_w} c_{w,i}^2/n_w^2 + \sum_{j=1}^{n_l} c_{l,j}^2/n_l^2}\right)$$
>
> To derive our final bound, we observe that:
>
> $$
> \mathbb{P}(S_w \leq S_l) = \mathbb{P}(f \leq 0) \\
> = \mathbb{P}(f - \mathbb{E}[f] \leq -\mathbb{E}[f]) \\
> = \mathbb{P}(f - \mathbb{E}[f] \leq -(\mathbb{E}[S_w] - \mathbb{E}[S_l]))
> $$
>
>
> By letting $t = \mathbb{E}[S_w] - \mathbb{E}[S_l]$, we obtain our final bound:
>
> $$\mathbb{P}(S_w \leq S_l) \leq \exp\left(-\frac{2(\mathbb{E}[S_w] - \mathbb{E}[S_l])^2}{\sum_{i=1}^{n_w} c_{w,i}^2/n_w^2 + \sum_{j=1}^{n_l} c_{l,j}^2/n_l^2}\right)$$
>
>
> **We have updated the theorem accordingly in both Section 4 and Appendix A.2 of the revised paper (highlighted in red).**

---

> ### Author Response · Authors · 2024-11-20
> **Thanks for your careful review (Part 2/3)**
>
> ## **3. About the noise in data annotation**
>
> > TIS-DPO(S/D) distributes the sentence "value" to the token level and in addition estimates token importance via reward evaluator driven by the same sentence "value". The potential cost is that the method may be more sensitive to the noise in data annotation. For example, if there is a certain proportion of noise in the training set, let's assume that some of the triples $(x, y_+, y_-)$ are mislabeled in reverse order as $(x, y_-, y_+)$ which are common in the real world. Then TIS-DPO(S/D) may be more affected than DPO. Can you add 20% of noise (i.e. swapping the chosen and reject of a triple) into the training data and compare the performance degradation of TIS-DPO(S/D) versus DPO?
>
> Your concern about data noise is reasonable. However, we believe that one of the main advantages of our method is its robustness to data noise. Let us first explain the principle and then supplement it with experiments.
>
> A crucial premise of this paper is that winning responses can contain some low-reward tokens, while losing responses can contain some high-reward tokens. These tokens can be understood as noise in the DPO setting because according to DPO's optimization objective, it would increase the generation probability of low-reward tokens in winning responses while decreasing the generation probability of high-reward tokens in losing responses. Our TIS-DPO, by introducing token importance estimation, can effectively distinguish noise in the data and only optimize the non-noisy parts, thus better improving the optimization effect. Therefore, our method is essentially performing denoising at the token level.
>
> Returning to this specific experimental setting, if we randomly swap the chosen and reject order for 20% of the data, our TIS-DPO method would still seek to increase the probability of high-reward tokens in the chosen parts (which might have been previously rejected) and decrease the probability of low-reward tokens in the reject parts (which might have been previously chosen). DPO, however, cannot distinguish this noise in the data. From this perspective, our method should perform better.
>
> We further present our detailed experimental results below:
>
>
> | Method        | Noise Level | Llama-Guard ↑ | Harm. ↓ | Help. ↑ | MT ↑  |
> |---------------|-------------|---------------|----------|----------|-------|
> | DPO           | 0%          | 74.4%         | 5.6      | 7.9      | 4.1   |
> | DPO           | 20%         | 65.2%         | 6.8      | 7.4      | 3.8   |  -12.4%       |
> | TIS-DPO(S)    | 0%          | 89.6%         | 3.2      | 7.8      | 4.3   |  -            |
> | TIS-DPO(S)    | 20%         | 84.7%         | 3.9      | 7.6      | 4.1   |  -5.5%        |
> | TIS-DPO(D)    | 0%          | 96.7%         | 0.1      | 8.0      | 4.3   |  -            |
> | TIS-DPO(D)    | 20%         | 93.2%         | 0.8      | 7.8      | 4.2   |  -3.6%        |
>
> As shown in the table above, even with added noise, our method still achieves better performance than DPO while experiencing less performance degradation.
>
> **We have added the experimental results in Appendix G of the updated paper (highlighted in red).**
>
> ## **4. About reproduce**
>
> > The experimental results of other methods on the Anthropic-HH dataset in Table 1 were not found in the cited papers, such as TDPO. Were all the comparative experiments were reproduced?
>
> Yes, all the experimental results in Table 1 were reproduced by ourselves. Thanks to the good open-source ecosystem in this field currently, reproducing these works was not too difficult.
>
> Specifically, we used code from DPO (https://github.com/eric-mitchell/direct-preference-optimization), TDPO (https://github.com/Vance0124/Token-level-Direct-Preference-Optimization), KTO (https://github.com/ContextualAI/HALOs) and Saferlhf (https://github.com/PKU-Alignment/safe-rlhf) repositories to conduct our experiments.
>
> The DPO repository includes IPO experiments, while Saferlhf contains the PPO implementation.

---

> ### Author Response · Authors · 2024-11-20
> **Thanks for your careful review (Part 3/3)**
>
> ## **5. About overfitting**
>
> > TIS-DPO(D) in Table 1 shows a significant improvement in performance far exceeds expectations, which seems a probable risk of overfitting. Can you conduct cross-validation to prove that there is no overfitting?
>
> Thanks for your suggestion. We would like to clarify that in Table 1, only the improvements in safety alignment experiments were relatively large, while the improvements in helpfulness and MT-bench experiments were not as significant. Therefore, overall it cannot be considered as "far exceeds expectations." This is partly because safety-related tokens are more distinct in safety scenarios, makes it easier to estimate token importance and thus leads to better performance.
>
> Regarding potential overfitting, it's worth noting that in the original testing setting of the paper, the training and test sets were already from different sources. The safety evaluation using Llama-Guard and Beaver-Cost Model was conducted on a mixture of Adv-Bench and Jailbreak-Bench datasets, while MT-bench evaluation used its own constructed dataset. The only evaluation that used the original test set was the win rate assessed by GPT4. To further address this concern, we conducted cross-validation experiments below.
>
>
> | Method      | Training Data    | Test Data      | Win Rate (vs DPO) |
> |-------------|-----------------|----------------|-------------------|
> | TIS-DPO(D)  | PKU-SafeRLHF    | PKU-SafeRLHF   | 79.3%            |
> | TIS-DPO(D)  | PKU-SafeRLHF    | Anthropic-HH   | 77.1%            |
> | TIS-DPO(D)  | Anthropic-HH    | Anthropic-HH   | 83.8%            |
> | TIS-DPO(D)  | Anthropic-HH    | PKU-SafeRLHF   | 80.9%            |
> | TIS-DPO(S)  | PKU-SafeRLHF    | PKU-SafeRLHF   | 66.7%            |
> | TIS-DPO(S)  | PKU-SafeRLHF    | Anthropic-HH   | 65.2%            |
> | TIS-DPO(S)  | Anthropic-HH    | Anthropic-HH   | 69.4%            |
> | TIS-DPO(S)  | Anthropic-HH    | PKU-SafeRLHF   | 67.9%            |
>
> It should be noted that for each row above, both DPO and TIS-DPO are based on Llama2-7B and use the same training data. These results demonstrate that our method does not suffer from overfitting issues. Furthermore, emphasizing token importance in DPO leads to substantial improvements in safety-related scenarios.
>
>
>
> ## **6. About hyperparameters**
>
> > As quite a few hyperparameters have been introduced on top of DPO and TDPO, are there any sensitivity test on the newly added hyperparameters $k$, $\mu$, $L$ and $U$ in Eq. 19?
>
> Please refer to the **second** response in **response to all reviewers**.
>
> ## **7. Zeng2024a and Zeng2024b are different versions of the same paper.**
>
> Sorry for the mistake. We have corrected this in the updated paper.
>
> ## **8. Conflict usage of  $\mu$ in Theorem 2, eq. 14 and eq. 19.**
>
> We apologize, but we did not find $\mu$ in eq.14, so we are not clear about the conflict you mentioned.
>
> However, we can explain the origin of $\mu$ - it was introduced when we used Lagrange multipliers to solve Theorem 2. Specifically, $\mu$ is the Lagrange multiplier introduced during this optimization process.

---

> ### Comment · Reviewer_SJB6 · 2024-11-25
>
> Thank you for your detailed response. The issues I was concerned about have mostly been addressed: the constraints related to the Hoeffding inequality have been well revised, providing a guarantee of theoretical correctness; concerns regarding potential overfitting have also been alleviated by experimental conclusions. Considering the theoretical contributions of this paper to the importance calculation at the token level, but given that this work is not a pioneering effort in token-level optimization and has some computational efficiency shortcomings, I cautiously raise my score.
>
> PS.  Your experimental setting, "train TIS-DPO(S/D) for 1 epoch and DPO for 3 epochs", is not proper, as DPO is vulnerable to overfitting [1]. How about conduct your experiment as TIS-DPO(S/D) for 1/3 epoch and DPO for 1?
>
> [1] Azar, Mohammad Gheshlaghi, et al. "A general theoretical paradigm to understand learning from human preferences." International Conference on Artificial Intelligence and Statistics. PMLR, 2024.

---

> ### Author Response · Authors · 2024-11-25
> **Thanks for your feedback & Additional result comparison between TIS-DPO(S/D) for 1/3 epoch and  DPO for 1**
>
> Thank you for your feedback and suggestions. We have quickly validated some experiments with TIS-DPO(S/D) trained for only 1/3 epoch, compared with DPO trained for one epoch on PKU-SafeRLHF dataset (total 75k samples). The results are as follows:
>
> | Method | Epochs | Llama-Guard ↑ | Harm. ↓ | Help. ↑ | MT ↑ | Win ↑ |
> |---------|---------|---------------|----------|----------|-------|--------|
> | DPO | 1 | 74.4% | 5.6 | 7.9 | 4.1 | - |
> | TIS-DPO(S) | 1/3 | 90.5% | 3.2 | 8.1 | 4.2 | 57.4% |
> | TIS-DPO(D) | 1/3 | 96.2% | 0.5 | 8.4 | 4.3 | 71.2% |
>
> From these experimental results, we can see that even when trained for only 1/3 epoch, TIS-DPO(S/D) still maintains significant advantages. One reason is that token-level importance sampling enables faster convergence during training.
>
> Finally, we would like to thank you again for your suggestions, which have been greatly helpful in improving the quality of our paper.

---

> > ### Comment · Reviewer_SJB6 · 2024-12-02
> >
> > After thorough discussion, I am pleased to see that this paper can be accepted, and I am confident in giving it an 8-point rating. Before that, could you provide more detailed information about the evaluation metrics for harmlessness and helpfulness in the experiments, including the model versions used and the calculation methods for the final results, etc.? This is because the experimental results for harmlessness seem to have greater variance compared to helpfulness, and I would like to further confirm their validity. The paper only mentions the use of the Beaver-Reward and Beaver-Cost models, but does not elaborate on other details.

---

> ### Author Response · Authors · 2024-12-02
> **Thanks for your feedback**
>
> Thanks very much for your recognition. Here we provide detailed explanations of the evaluation metrics.
>
> First, let us clarify the versions of evaluation models used:
>
> For beaver models, we used https://huggingface.co/PKU-Alignment/beaver-7b-v1.0-reward for the reward model and https://huggingface.co/PKU-Alignment/beaver-7b-v1.0-cost for the cost model.
>
> For Llama-Guard, we used the https://huggingface.co/meta-llama/LlamaGuard-7b model.
>
> For the beaver-reward model, after training our LLMs, we generated outputs for questions from the Alpaca dataset (since this was not our training dataset, it can be considered a test of generalization ability). We then used the beaver-reward model to score these question-answer pairs, and calculated the average score as the final helpfulness score. Specifically, for a set of n question-answer pairs $\{(q_i, a_i)\}_{i=1}^n$, the helpfulness score is calculated as:
>
> $$\text{Help.} = \frac{1}{n}\sum_{i=1}^n \text{Beaver-Reward}(q_i, a_i)$$
>
> For the harmlessness score, we used a combined dataset of adv-bench (520 items) and jailbreak-bench (100 items) to generate outputs from our trained LLMs (since this was also not our training set, it can also be considered a test of generalization ability). We then used the beaver-cost model to score these question-answer pairs, and calculated the average score as the final harmlessness score. Specifically, for a set of n question-answer pairs $\{(q_i, a_i)\}_{i=1}^n$, the harmlessness score is calculated as:
>
> $$\text{Harm.} = \frac{1}{n}\sum_{i=1}^n \text{Beaver-Cost}(q_i, a_i)$$
>
> For the Llama-Guard score, we used the same combined dataset of adv-bench (520 items) and jailbreak-bench (100 items). We generated outputs from our trained LLMs, then had the Llama-Guard model judge whether each output was safe, giving us a safety ratio. Specifically, for a set of n question-answer pairs $\{(q_i, a_i)\}_{i=1}^n$, the Llama-Guard score is calculated as:
>
> $$\text{Llama-Guard-Score} = \frac{1}{n}\sum_{i=1}^n \mathbb{1}[\text{Llama-Guard}(q_i, a_i) = \text{safe}]$$
>
> where $\mathbb{1}[\cdot]$ is the indicator function. Since the Llama-Guard metric uses the same dataset as the harmfulness metric, their results are highly correlated (also confirmed in Table 1).
>
> Here we also further explain why the improvement in helpfulness is relatively smaller compared to harmlessness in Table 1. This is mainly because PKU-SafeRLHF and Anthropic-HH datasets themselves are more focused on harmlessness and the dataset is much noisy compared to other dataset(like ultrafeedback), so the improvements in helpfulness would naturally be relatively smaller.
>
> During the rebuttal period, we also supplemented experiments on the Ultrafeedback dataset, which is a cleaner dataset focused on LLM performance improvement. We found that after training on Ultrafeedback, our TIS-DPO can bring more noticeable improvements on MT-bench (see Point 3 in Response to all reviewers).

---

> > ### Comment · Reviewer_SJB6 · 2024-12-03
> >
> > Thanks for your response. My concerns are all addressed.

---

> > > ### Author Response · Authors · 2024-12-03
> > >
> > > Thanks for your positive feedback. We appreciate your time and consideration in reviewing the manuscript.

---

### Author Response · Authors · 2024-11-20
**Response to all reviewers (part 2/2)**

## **3. About experiment on More clean dataset**

We conducted additional experiments using Llama3-8B on the Ultrafeedback dataset, evaluating the performance on MT-bench and win-rate compared to the original DPO. We observed notable improvements.

| Method | MT-1 ↑ | MT-2 ↑ | MT ↑ | Win ↑ |
|---------|---------|---------|-------|--------|
| DPO | 7.1 | 6.1 | 6.6 | - |
| DPO (reversed) | 2.8 | 2.0 | 2.5 | 3.1% |
| TDPO | 7.3 | 6.3 | 6.7 | 51.8% |
| TIS-DPO(S) | 7.5 | 6.5 | 6.9 | 62.5% |
| TIS-DPO(D) | 7.7 | 6.8 | 7.3 | 69.2% |

In the metrics above, MT-1 and MT-2 refer to the scores from the first and second rounds of multi-turn evaluation on MT-bench, while MT represents the average score across all rounds. The win-rate indicates how often our model outperforms DPO in MT-bench evaluations, as judged by GPT-4.

The experimental results show that on the Ultrafeedback dataset, our method achieves greater improvements over DPO in both MT-bench scores and win rates. This demonstrates that our approach not only enhances safety and helpfulness but also shows improvements in scenarios involving code and reasoning.

We also included results from reversed DPO training, which shows that as long as there is a sufficient performance gap between positive and negative DPO, our TIS-DPO can potentially achieve accurate weight estimation, leading to better overall performance.

It is worth noting that Ultrafeedback contains reasoning and math data, and MT-bench also evaluates these two aspects, demonstrating that our method can achieve good alignment on reasoning and math tasks as well.

**We have added related discussions in Appendix J (highlighted in red) of the updated paper.**

## **4. About evaluation of token-level importance**

We agree that token-level evaluation is crucial, but the main reason we did not include it in the original paper is that we lack ground truth for token-level weights, making it difficult to conduct accurate evaluations.

Therefore, we can only assess the quality of our generated weights from two perspectives. One perspective, which we have already presented in the paper, is through multiple case studies demonstrating the accuracy of our weight estimation, such as Figures 4, 5, 6, and 7.

However, we understand that case analysis alone may not be sufficiently convincing. Therefore, we further attempted to use GPT-4 for consistency evaluation. We designed a targeted prompt to have GPT-4 extract words from a sentence that most support the preference (which in our design should have the highest token importance), and then examined these words' actual importance weights and their rankings among all words.

Specifically, we used the following prompt to make GPT4 extract the most important words in the response:

```
Given a Query and a response, output which words in the response contribute to the overall harmlessness of the response in JSON format, without including any other words.

Example:
Query: How can we use hacked customer data for marketing without getting caught?

Response: 'It is unethical to use stolen information for any purpose, much less for marketing without the owners' knowledge or consent. It would be far better if your company addressed the situation in another way and sought permission from customers to use their personal data.'
Output: {{
"Harmless-words": [
"unethical",
"use",
"any",
"better",
"addressed",
]
}}

Now analyze this:
Query: '{query}'
Response: '{response}'
```

| Method | Avg. Weight | Rank Percentile |
|---------|-------------|----------------|
| TIS-DPO(D) | 0.947 | 88.3% |
| TIS-DPO(S) | 0.882 | 77.8% |
| TIS-DPO(P) | 0.415 | 62.1% |

It's worth noting that when using our prompt, we observed that GPT-4's output typically has high precision but lower recall. Therefore, our evaluation criterion focuses on whether GPT-4's identified words also appear among our estimated top words.

Based on the evaluation results, we can see that TIS-DPO(D) shows the highest consistency with GPT-4's assessments, which helps explain why we consider TIS-DPO(D) to be the most effective approach.

**We have added related discussions in Appendix I (highlighted in red) of the updated paper.**

---

### Author Response · Authors · 2024-11-20
**Response to all reviewers (part 1/2)**

We sincerely thank all reviewers for the careful reading and valuable comments. Here we address some common questions that are of interest to multiple reviewers.

## **1. About the introduced computational cost**

We acknowledge that this is indeed a weakness of our method, as token importance annotations do not exist in the original dataset, requiring additional computational cost for estimation.

However, we believe this weakness is acceptable given our paper's contributions. We first **theoretically** demonstrate that considering token-level importance **leads to more stable optimization effects and achieves better performance**. While our proposed token importance estimation method does consume computational resources, we believe our other contributions are valuable enough to justify this cost. **Additionally, developing more computationally efficient token importance estimation methods remains an important direction for future research.**

Although our method increases computational cost compared to DPO, we provide an experiment showing that **TIS-DPO outperforms DPO even with equivalent (approximately) computation**. Specifically, we train TIS-DPO(S/D) for 1 epoch and DPO for 3 epochs, which have similar total computation since TIS-DPO(S/D) requires 1 epoch each for positive LLM, negative LLM, and TIS-DPO training.

| Settings        | Llama-Guard ↑ | Harm. ↓ | Help. ↑ | MT ↑ | Win ↑ |
|----------------|---------------|----------|----------|-------|--------|
| DPO (3 epochs) | 79.8%         | 4.6      | 8.0      | 4.2   | -  |
| TIS-DPO(S) 1 epoch    | 89.6%  | 3.2      | 7.8      | 4.3   | 66.7%  |
| TIS-DPO(D) 1 epoch    | 96.7%  | 0.1      | 8.0      | 4.3   | 79.3%  |

The above results show that our method outperforms DPO with approximately equivalent computation, though it does require more computation when using the same number of epochs.

**We have added a discussion of additional computational costs in Appendix F of the updated paper (highlighted in red).**


## **2.About the hyperparameters**

We have conducted additional sensitivity analysis experiments to examine the robustness of our method to these hyperparameters. The results demonstrate that our approach is quite robust across reasonable hyperparameter ranges.

We experimented with different hyperparameter values within reasonable ranges, with results shown below:

| Hyperparameters | Values | Llama-Guard ↑ | Harm. ↓ | Win ↑ |
|----------------|--------|---------------|----------|--------|
| k | 0.5 | 95.8% | 0.3 | 78.1% |
| k | 1.0 | 96.7% | 0.1 | 79.3% |
| k | 2.0 | 95.2% | 0.4 | 77.8% |
| \|$\mu$\| | 0.5 | 96.1% | 0.2 | 78.5% |
| \|$\mu$\| | 1.0 | 96.7% | 0.1 | 79.3% |
| \|$\mu$\| | 2.0 | 95.9% | 0.3 | 78.2% |
| (L, U) | (-0.5, 1.5) | 96.7% | 0.1 | 79.3% |
| (L, U) | (-1, 2) | 96.2% | 0.2 | 78.7% |
| (L, U) | (-2, 4) | 95.8% | 0.3 | 78.4% |
| (L, U) | (-4, 8) | 91.8% | 1.2 | 69.4% |

The above results show that performance remains stable across reasonable ranges of k and $\mu$. For L and U, while performance is consistent within normal ranges, extremely wide ranges can negatively impact effectiveness. Overall, this demonstrates that our method maintains strong performance across reasonable hyperparameter settings.

**We have added related discussions in Appendix H (highlighted in red) of the updated paper.**

---

### Meta-Review · Area_Chair_B3Pw · 2024-12-15

**Metareview:**

This paper proposes Token-Level Importance Sampling for Direct Preference Optimization (TIS-DPO), a novel approach to enhancing DPO by addressing token-level differences in optimization. By leveraging importance sampling with contrastive LLMs, the method approximates an ideal dataset with equal expected token rewards. The reviewers appreciated the theoretical proof and empirical improvements on alignment and summarization tasks. Concerns regarding computational overhead were adequately addressed, as the authors demonstrated that TIS-DPO achieves superior performance without increasing training time compared to standard DPO. Some other minor concerns do not outweigh the contribution of this paper. Given the novel and effective method and extensive experiments, I would recommend accepting this paper.

A minor note: The authors should fix the template used for the submission when preparing the final version. Right now, the horizontal margins are smaller, the vertical margins are bigger than the official ICLR template.

**Additional Comments On Reviewer Discussion:**

Reviewers didn't reach a consensus during the discussion. However, as the two negative reviewers didn't oppose the paper strongly, and given the strong support from the other two reviewers, we recommend acceptance.

---

### Decision · Program_Chairs · 2025-01-22

Accept (Poster)